# Piperlongumine as a Neuro-Protectant in Chemotherapy Induced Cognitive Impairment

**DOI:** 10.3390/ijms23042008

**Published:** 2022-02-11

**Authors:** Fabio Ntagwabira, Madison Trujillo, Taylor McElroy, Taurean Brown, Pilar Simmons, Delawerence Sykes, Antiño R. Allen

**Affiliations:** 1Division of Radiation Health, University of Arkansas for Medical Sciences, 4301 W. Markham Street, Little Rock, AR 72205, USA; fntagwabira@uams.edu (F.N.); mtrujillo@uams.edu (M.T.); tmmcelroy@uams.edu (T.M.); tbrown8@uams.edu (T.B.); pgsimmons@uams.edu (P.S.); 2Department of Pharmaceutical Sciences, University of Arkansas for Medical Sciences, 4301 W. Markham Street, Little Rock, AR 72205, USA; 3Neurobiology & Developmental Sciences, University of Arkansas for Medical Sciences, 4301 W. Markham Street, Little Rock, AR 72205, USA; 4Department of Biology, Pomona College, Claremont, CA 91711, USA; delawrence.sykes@pomona.edu

**Keywords:** brain, memory, chemotherapy, piperlongumine

## Abstract

Advances in the early diagnosis and treatment have led to increases in breast cancer survivorship. Survivors report cognitive impairment symptoms such as loss of concentration and learning and memory deficits which significantly reduce the patient’s quality of life. Additional therapies are needed to prevent these side effects and, the precise mechanisms of action responsible are not fully elucidated. However, increasing evidence points toward the use of neuroprotective compounds with antioxidants and anti-inflammatory properties as tools for conserving learning and memory. Here, we examine the ability of piperlongumine (PL), an alkaloid known to have anti-inflammatory and antioxidant effects, to play a neuroprotective role in 16-week-old female C57BL/6J mice treated with a common breast cancer regimen of doxorubicin, cyclophosphamide, and docetaxel (TAC). During social memory testing, TAC-treated mice exhibited impairment, while TAC/PL co-treated mice did not exhibit measurable social memory deficits. Proteomics analysis showed ERK1/2 signaling is involved in TAC and TAC/PL co-treatment. Reduced Nrf2 mRNA expression was also observed. mRNA levels of Gria2 were increased in TAC treated mice and reduced in TAC/PL co-treated mice. In this study, PL protects against social memory impairment when co-administered with TAC via multifactorial mechanisms involving oxidative stress and synaptic plasticity.

## 1. Introduction

Breast cancer is the most common invasive cancer among women worldwide. In the United States, breast cancer is the leading cause of cancer death among women aged 20–59 years. Despite high incidence rates, advances in diagnostic and treatment methods have led to an increase in survivorship [1]. However, cancer survivors often experience chemotherapy-induced side effects such as fatigue, difficulty concentrating, learning difficulties, and loss of memory that can negatively impact the quality of life. These symptoms are collectively referred to as chemotherapy-induced cognitive impairment (CICI). The need for therapies that can prevent or alleviate these side effects is thus crucial. 

Currently, a commonly used anthracycline-based adjuvant treatment for breast cancer is doxorubicin and cyclophosphamide followed by docetaxel (TAC), which has been linked to improved disease-free survival and overall survival rates. Current uses of doxorubicin, a derivative from the Streptomyces peucetius bacterium, include treatment of breast, ovary, bladder, and thyroid cancers. It intercalates with DNA base pairs, causing DNA strand breaks, thereby inhibiting DNA and RNA synthesis [2]. Cyclophosphamide was FDA approved as an anti-cancer agent in 1959. The mechanism of action involves its break down to phosphoramide metabolite, which causes cell death by cross-linking adjacent DNA strands at the guanine N-7 [3]. Docetaxel, a derivative of extracts from leaves of European yew tree Taxus baccata [4], was granted FDA approval in 1996 for use against metastatic breast cancer, and it is currently used to treat breast, gastric, head and neck, ovary, and prostate cancers [5]. Its main mechanism of action involves cell cycle arrest by stabilizing beta-tubulin [6]. These chemotherapeutic agents induce neurotoxicity in patients. In fact, cyclophosphamide and doxorubicin have been linked to verbal memory problems in breast cancer patients [7], and docetaxel has been shown to affect the quality of life in lung cancer patients [8]. Although the CICI mechanism of action has not been deduced, contributing factors include inflammation, senescence, and oxidative stress [9]. Hence, compounds with antioxidant, anti-inflammatory, and anti-cancer effects are promising candidates to combine with chemotherapy to combat CICI.

PL is an alkaloid component of the long pepper *Piper longum* L. used in traditional Ayuverdic medicine in Latin America and India. Since it was discovered and characterized in the 1960s [10], the therapeutic potential of PL has been extensively explored in vivo and in vitro. PL yields anti-tumor anti-cancer properties but may also function as an antidepressant and anxiolytic [11]. PL may be used to selectively target cancer cells while minimally affecting normal cells [12]. Furthermore, PL is one of several molecules that has been characterized to have the ability to selectively kill senescent cells with a low toxicity profile as an advantage compared to other senolytics [13]. The mechanisms of action that have been documented to date show that PL works by targeting several molecular mechanisms involved in cancer, such as phosphatidylinositol 3-kinase/protein kinase B (PI3K/Akt), nuclear factor kappa B (NF-κB), and cyclooxygenase-2 (COX-2) [14]. The main challenge for using PL is identifying specific molecular targets. However, due to the lack of a known specific mechanism of action, this remains a difficult task [15]. The use of PL as a co-therapeutic improves cognition in an Alzheimer’s mouse model and aged mice [16,17] but, to our knowledge, recent empirical evidence examining the role of PL against TAC induced CICI is limited. Here we use a combined treatment of TAC and PL to evaluate its impact on hippocampal-dependent social memory in C57BL/6J female mice.

## 2. Results

### 2.1. Food Consumption and Body Weight

Food consumption and body weight were monitored during the period of chemotherapy injections. Overall, neither variable differed significantly as a result of the treatment group. Significant differences relative to control mice were observed, however, in treatment-by-week interactions for body weights of mice treated with PL in week 6 F [7.101,78.11] = 2.201, *p* < 0.05) (Figure 1A); Bonferroni’s multiple comparison revealed TAC/PL treatment significantly decreased body weight 6 weeks post-treatment compared to DMSO alone. For food consumption, there were significant differences between treatment-by-week interaction F [1.817,3.633] = 11.61, *p* < 0.05) (Figure 1B); Bonferroni’s multiple comparison revealed TAC/PL significantly decrease food consumption week 2 and TAC/DMSO week 6 compared to DMSO only.

### 2.2. Three-Chamber Sociability

Social behavior was evaluated with the 3-chamber social approach [18]. During the habituation stage, all mice spent approximately equal time exploring both lateral chambers (Figure 2A; One-way ANOVA, F [7,72] = 1.67, *p* = 0.13). Normal social behavior was observed during the sociability phase, when all animals spent significantly more time exploring the chamber where the first newly introduced mouse (stimulus 1) was located (Figure 2B; One-way ANOVA, F [7,72] = 11.33, *p* < 0.0001 Stimulus vs. Empty). TAC affected social discrimination during the third stage where chemotherapy treatment elicited an inability to discriminate between the familiar and stranger mouse, while DMSO, PL, and TAC/PL animals successfully spent more time exploring the novel stranger (Figure 2C; One-way ANOVA, F [7,72] = 3.006, *p* = 0.0079).

### 2.3. Proteomics

#### 2.3.1. Protein Numbers Differentially Expressed in Association with Chemotherapy Induced Cognitive Impairment and Treatment with PL

A total of 4769 proteins were identified in the TAC-treated and TAC/PL-treated mice. Relative to control, TAC-treated and TAC/PL-treated groups had 143 and 123 differentially expressed proteins, respectively. Table 1 and Table 2 show the top dysregulated proteins affected with TAC treatment and TAC/PL co-treatment, respectively.

#### 2.3.2. Identification of Proteomics Pathways and Networks Involved Chemotherapy-Induced Cognitive Impairment and Treatment with PL 

IPA, a web-based application used to visualize, analyze, and understand “-omics” data, was used to identify proteomic pathways involved in PL treatment against chemotherapy-induced cognitive impairment. IPA computes the ratio between the numbers of molecules that meet the specified criteria against the total number of proteins in the IPA database; the *p*-value indicates that ratio occurs by chance.

IPA indicated differentially expressed proteins and the networks associated with TAC treatment relative to control (Table 3, Figure 3). The identified functions are associated with DNA damage and repair, cell signaling, cell metabolism, and cancer pathways. EIF2 signaling, mitochondrial dysfunction, and oxidative stress were identified among the top canonical pathways in relation to TAC treatment (Table 4, Figure 4). Oxidative stress and mitochondrial dysfunction were previously reported to be involved in multiple models of cognitive impairment, including chemotherapy-induced cognitive impairment [19]. With the same approach, we identified protein networks and pathways associated with TAC, compared to TAC/PL treatments (Table 5, Figure 5). The identified functions include post-translational modification, nervous system development and function, cell morphology, cell development, and cell function. The top canonical pathways related to PL treatment include nerve growth factor (NGF) signaling, apelin muscle signaling, and extracellular signal-regulated kinase (ERK) 5 signaling (Table 6, Figure 6).

### 2.4. Changes in mRNA Expression

#### 2.4.1. Nrf2 Pathway Molecules 

We first evaluated the effects of our treatment regimen on expression of nuclear factor erythroid 2-related factor 2 (Nrf2). Treatment with PL, TAC, and TAC/PL reduced Nrf2 mRNA levels (Figure 7A; One-way ANOVA, F [3,26] = 10.23, *p* = 0.0001). Bonferroni multiple comparisons indicate a decrease from DMSO to PL (*p* < 0.05), DMSO to TAC (*p* < 0.001) and DMSO to TAC/PL (*p* < 0.001). We evaluated the effects of our treatment regimen on mRNA levels of antioxidant response element (ARE) genes that are affected by Nrf2 activation. Expression of the glutamate cysteine ligase enzyme modifier subunit (GCLM) was lower in all treatment groups than in the control per (Figure 7B; One-way ANOVA, F [3,25] = 10.13, *p* = 0.0002). Bonferroni multiple comparisons indicate a decrease from DMSO to TC (*p* < 0.001) and DMSO to TAC/PL (*p* < 0.01). The glutamate cysteine ligase enzyme catalytic subunit (GCLC) was differentially expressed between treatment groups (Figure 7C; One-way ANOVA, F [3,23] = 19.89, *p* < 0.0001). The analysis also revealed that treatment groups had significant differences in expression of NADPH quinone dehydrogenase 1 (Nqo1) (Figure 7D; One-way ANOVA, F [3,26] = 9.558, *p* = 0.0002). Bonferroni multiple comparisons indicate an increase from DMSO to PL (*p* < 0.01). Heme oxygenase 1 (Hmox1) (Figure 7E; One-way ANOVA, F [3,26] = 10.02, *p* = 0.0001). 

Bonferroni multiple comparisons indicate a significant decrease from DMSO to PL (*p* < 0.001) and DMSO to TAC/PL (*p* < 0.001). Finally, we examined thioredoxin reductase 1 (Txnrd1) (One-way ANOVA, F [3,24] = 10.01, *p* = 0.0002), Bonferroni multiple comparisons indicate a decrease from DMSO to PL (*p* < 0.05), DMSO to PL (*p* < 0.001) and DMSO to TAC/PL (*p* < 0.001).

#### 2.4.2. NMDA/AMPA Gene Coding Subunits 

We evaluated changes in mRNA expression of NMDA subunits (Grin1, Grin2a, Grin2b) and AMPA subunits (Gria1, Gria2). mRNA levels of NMDA subunit Grin1 did not change in response to any of the treatments (One-way ANOVA, F [3,26] = 1.855, *p* = 0.1620). Animals treated with TAC/PL underwent significant Grin2a down-regulation (Figure 8A; One-way ANOVA, F [3,26] = 6.177, *p* = 0.0026. Bonferroni multiple comparisons indicate a decrease from DMSO (*p* < 0.05). PL only and TAC/PL treatment significantly down regulated Grin2b (Figure 8B; One-way ANOVA, F [3,26] = 21.22, *p* < 0.0001). Bonferroni multiple comparisons indicate a decrease from DMSO to PL (*p* < 0.05) and from DMSO to TAC/PL (*p* < 0.001). mRNA levels of AMPA subunit Gria1 did not undergo changes in response to the treatments (One-way ANOVA, F [3,26] = 2.086, *p* = 0.1266). PL and TAC treatment increased Gria2 mRNA levels (Figure 8C; One-way ANOVA, F [3,26] = 6.966, *p* = 0.0014). Bonferroni multiple comparisons indicate a decrease from DMSO to PL (*p* < 0.05) and from DMSO to TAC (*p* < 0.01).

## 3. Discussion

In the current study of C57BL/6 mice, we investigated how a combination chemotherapy regimen (TAC) affected social memory (i.e., the ability to recognize and remember members of the same species), and we examined the ability of PL to improve deficits in social memory. TAC treatment in mice previously was shown to induce impairment of hippocampal-dependent spatial memory [20] and deficits in hippocampal-dependent short-term memory [21]. In mice, social memory can be a powerful tool for exploring hippocampal learning and memory because it displays similar features to other forms of hippocampal-dependent memories [22].

During the habituation phase, as animals freely explored the left and right compartments, the amount of time they spent exploring each did not significantly differ due to the lack of a conspecific in either chamber. During the sociability stage, when an unfamiliar mouse is introduced into one of the chambers and an empty cage remains in the other, mice spent significantly more time with the conspecific than with the inanimate object because mice are social animals [23]; we observed the same trend in all treatment groups. 

We assessed social novelty (i.e., the propensity for a mouse to spend more time with a novel mouse instead of a familiar mouse when presented with both choices simultaneously) [23] to investigate social memory, which is required for a mouse to distinguish between a familiar mouse and a novel mouse. During the social novelty stage of our study, TAC-treated mice displayed no significant differences in the amount of time spent exploring the familiar or novel mouse, suggesting that mice in this group had impaired social novelty that prevented them from distinguishing the familiar mouse from the novel mouse. Animals treated with both TAC and PL, however, appeared to have intact social memory; they spent significantly more time with the novel mouse than with the familiar mouse, perhaps due to PL incorporation. In addition, mice that received only control (DMSO) or PL treatment displayed normal social novelty, indicating no impact on social memory. Although PL’s ability to improve cognition is not fully understood, our results are in agreement with previous findings that PL reduces age-related cognitive decline and improves hippocampal neurogenesis in mice and that it protects against Alzheimer’s disease pathophysiology in hippocampal neurons of a mouse model [17].

We used a bottom-up proteomics approach to examine how TAC treatment affects the hippocampus in female mice. The identified upregulated proteins included DBI, which has been shown to be upregulated in several human brain tumors [24]; cytochrome C, which plays an important role in apoptosis; and the NDFB3 complex, which is part of the mitochondrial membrane respiratory chain (Table 1). Mutations in the complex are associated with mitochondrial dysfunction and play key roles in neurodegenerative diseases [25]. In addition, NMT2 was downregulated. The NMT2 enzyme is crucial due to its role in co- or post-translation modification of several proteins that impact protein–protein interactions involved in several metabolic pathways [26]. The canonical pathways most affected by the chemotherapy regimen tested in our experiments included EIF2 signaling, oxidative phosphorylation, and mitochondrial dysfunction (Figure 4 & Table 4); all have been implicated in neurodegenerative diseases.

The top networks identified by IPA as affected by TAC treatment included proteins that are important for broad biological functions such as DNA-damage repair, nucleic acid metabolism, and protein synthesis (Table 3). At the center of network 1 lies LARP7, a transcription regulator [27] whose expression levels play a role in cancer progression and metastasis [28]. Network 2 has several interactions with the antioxidant enzyme superoxide dismutase (SOD-1) (see Appendix A). Doxorubicin has been reported to cause cardiotoxicity by reducing SOD-1 expression levels due to increased oxidative stress [29]. Network 3 (Appendix A) highlights several interactions between proteins associated with the electron transport chain and mitochondrial function. The effects of doxorubicin on mitochondrial dysfunction have been reported [30]. The chemotherapeutic regimen used in the experiments is most likely the reason for the observed interactions in this network. The ERK (p44/44 mapk) pathway also appeared to be affected. Anthracyclines, including doxorubicin, have been reported to play roles in apoptosis, cell proliferation, and cell cycle progression [31]. Together, these observations might provide insight into cognitive impairment that results from TAC treatment. 

Proteomics was used to analyze the impact of co-treatment with PL and TAC (i.e., TAC/PL). The notable canonical pathways (Figure 6 & Table 6) included the NGF signaling pathway, which plays an important role in maintaining neuron populations in the nervous system by regulating cell death and survival. NGF belongs to a family of proteins known as neurotrophins, which are responsible for the development, survival, and function of neurons and have been associated with neurodegenerative disease states and symptoms [32]. Thus, the NGF signaling pathway may provide insight into the mechanisms underlying chemotherapy-induced cognitive impairment. Apelin muscle, ERK5, estrogen-dependent breast cancer, and hypoxia in the cardiovascular system signaling pathways were also affected (Figure 6 & Table 6). These pathways overlap in their abilities to affect several processes that include inflammation, cancer progression, cell survival, and neurodegeneration [33,34,35].

The top networks identified by IPA as affected by co-treatment with TAC and PL indicated functions such as post-translational modification, RNA post-transcriptional modification, and cellular development (Table 5). Network 1 mainly centers around ERK1/2 signaling (Figure 5), which regulates several cellular processes such as cell cycle, cell proliferation, cell differentiation, and apoptosis [36]. The pathways are evolutionarily conserved, and genetic and epigenetic mutations in molecules involved are implicated in several cancers, making them good therapeutic target candidates. For instance, doxorubicin is known to target cancer cells by inducing DNA damage through ERK activation [37]. In addition, PL reportedly elicits its anti-cancer effects by activating the MEK/ERK pathway [38]. 

Network 2 analysis indicated that TAC/PL co-treatment affected pathways involved in maintaining cellular function, development and function of the nervous system, and tissue development. A major focus molecule was CREB1 (Appendix A), which is a transcriptional factor for cell survival, cancer development, metastasis [39,40], and many other processes. CREB function is also important to normal neuronal function and the formation of long-term memory [41]. Disruptions in CREB expression levels have been implicated in cognitive impairment, with phosphorylated CREB levels being associated with good memory performance [42]. Go et al. reported that aged mice treated with PL had increased levels of phosphorylated CREB in the hippocampal CA3 region and better performance in behavior tests [16]. 

Network 3 showcased TRIM25 as one of the hub proteins in the interactions (Appendix A) of pathways affected by TAC/PL co-treatment. TRIM25 regulates endoplasmic reticulum (ER) stress by degrading misfolded proteins via ER-associated degradation [43]. Liu et al. recently reported that TRIM25 targets Keap 1 by ubiquitin-mediated degradation that leads to Nrf2 activation in hepatocellular carcinoma [44]. Peng et al. reported that PL and its analogs elicit neuroprotective effects against ROS through a similar method whereby PL disrupts Keap1 interactions with TRIM25, leading to activation of Nrf2 and expression of ARE genes [45]. Because oxidative stress plays a role in most neurodegenerative disease forms, including CICI [46], investigating molecules involved in the cellular antioxidant system is likely to provide insights into neurodegeneration caused by chemotherapy and possible pathways through which co-administration of PL may be neuroprotective against the observed effects of TAC treatment on social memory. 

We used qRT-PCR to determine the effects of our treatment regimen on the genes encoding Nrf2, AREs activated by Nrf2 and subunits of NMDA and AMPA. Nrf2 regulates several genes involved in protection from cellular oxidative stress, including in the central nervous system. Glutathione is the main cellular defense mechanism against oxidative stress. Its synthesis involves cysteine and the activity of glutamate cysteine ligase (GCL), which is composed of a regulatory subunit/modifier subunit (GCLM) and a catalytic subunit (GCLC). In our experiments, TAC treatment resulted in reduced expression of GCLM (Figure 7B) and GCLC, although the latter reduction was not statistically significant; this might be explained by similar patterns of Nrf2 expression. It appears that TAC/PL co-treatment contributes more to the expression of GCLC than of GCLM; treatment resulted in increased expression of GCLC and reduced expression of GCLM. Although GCLM confers some enzymatic activity to GCL, GCLM is the active subunit, facilitating glutathione synthesis by reducing the Km for glutamate and the Ki for glutathione [47]. Nrf2 activation also is required for the expression of Nqo1, which is a quinone reductase and superoxide reductase that plays several roles in cellular protection against oxidative stress [48]. Reduced activity and increased production of Nqo1 have been linked to doxorubicin treatment as markers of cardiotoxicity [49]. Downregulation of Nqo1 would be expected to correspond to the increased expression of Nrf2, but the opposite effect occurred in TAC-treated mice relative to control mice (Figure 7D). TAC/PL co-treatment, however, resulted in reduced expression of Nqo relative to results of TAC treatment. 

Recent studies have shown versatile roles for Nqo, and it is possible that its upregulation might not be related to Nrf2 [50]. TAC treatment resulted in reduced mRNA levels of Hmox1 (Figure 9E), which is protective against oxidative stress [51]. TAC and TAC/PL treatments reduced expression of Hmox1, and decreased expression of Hmox1 has been associated with age-related impaired memory [52]. A previous study in a model of Alzheimer’s disease showed that Hmox1 expression might not always be related to Nrf2 [53]. Our co-treatment did not appear to improve Hmox1 expression compared to TAC, but PL treatment alone did. Txnrd1 expression followed a similar pattern: PL, TAC, and TAC/PL treatments, compared to control, resulted in significantly reduced mRNA levels of Txnrd1, and effects of TAC and TAC/PL did not differ. Txnrd1 has many cellular functions, one of which is protecting against oxidative stress by acting as a reducing agent. Reduced mRNA expression of Txnrd1 is associated with Parkinson’s disease [54], which has a pathology similar to CICI due to oxidative stress.

Although several reports have directly linked increased or repressed expression of Nrf2 to expression of the antioxidant genes we examined, our treatment regimens resulted in different trends. The role of Nrf2 in oxidative stress has been well documented [55]. Traditionally, chemotherapy treatment increases oxidative stress [56], which should increase Nrf2 expression. Although that is not what we observed in this study, there is evidence that Nrf2 targets can be post-transcriptionally activated, independent of the expression levels of their target molecules. McElroy et al. reported post-transcriptional activation of GCL with no effects on the expression of Nrf2 and other target molecules [57,58].

NMDA and AMPA are ionotropic glutamate receptors implicated in long-term potentiation (LTP), which is necessary for learning and memory formation [59], as well as other functions of the central nervous system. Altered functioning of NMDA receptors has been linked to central nervous system pathologies that include Alzheimer’s disease, autism, and other neurodegenerative diseases. Although there are variations in the subunits that form the NMDA ion channel, those encoded by Grin1, Grin2a, and Grin2b typically are more prevalent [60]. Our treatments did not induce changes in mRNA levels of Grin1. The mRNA levels of Grin2b, however, increased in response to PL, which contradicts results from aged mice and a mouse model of Alzheimer’s disease [17]. Go et al. reported that PL treatment had no effect on Grin2b but improved cognition by increasing levels of phosphorylated NR2B (gene product of Grin2b), in addition to calmodulin-dependent protein kinase II alpha and ERK1/2 in aged mice [16]. The contradicting results could be due to differences in administration of PL—the previous study used oral gavage, but we used i.p. injection. Surprisingly, TAC/PL co-treatment significantly reduced mRNA levels of Grin2a and Grin2b, relative to control (Figure 8A,B), even though this combination improved TAC-induced impairment of social memory. Collectively, our observations suggest that the mechanism underlying PL protection against TAC-induced social memory impairment may act at the protein level of NMDA receptors or may involve deficits in other subunits not considered in this study. 

Of the genes encoding AMPA receptor subunits, our treatment regimen affected only Gria2, which encodes GluA2. AMPA receptors also play an important role in synaptic strength and plasticity, and deficits are associated with neurodegenerative pathologies. AMPA receptors consist of 4 subunits (GluA1-4), with GluA1/2 (encoded by Gria1 and Gria2, respectively) being the majority phenotype [61]. Both PL treatment and TAC treatment increased mRNA levels of Gria2, and TAC/PL had no effect on expression (Figure 8C). While we observed no changes in Gria1, increased levels of GluA2 have been reported to be associated with memory deficits after doxorubicin treatment [62]. In fact, GluA2 knockout mice displayed an increase in LTP [63], which agrees with our results. Although we observed no deficits in expression levels of the genes encoding NMDA receptor subunits, the resulting TAC-associated social memory impairment could be explained by the observed increase in mRNA levels of Gria2 (GluA2) in TAC-treated mice. TAC/PL co-treatment had no impact on mRNA expression levels of Gria2, suggesting a mechanism involving AMPA receptors.

## 4. Materials and Methods

### 4.1. Animals

Female C57BL/6 mice (Jackson Laboratory, Bar Harbor, ME, USA), 16 weeks old (*n* = 48), were used in this study. Mice were housed together according to the treatment group, with no more than 5 per cage. Food intake and weight were monitored weekly during the injection period, and water was provided as needed. Housing followed a constant 12-h light/12-h dark cycle. Animal experiments were conducted in compliance with the National Research Council’s Guide for the Care and Use of Laboratory Animals and the Institutional Animal Care and Use Committee at the University of Arkansas for Medical Sciences (UAMS).

### 4.2. Experiment Design and Chemotherapy Regimen 

Chemotherapeutic agents (docetaxel, doxorubicin, cyclophosphamide; TAC) and 0.9% sodium chloride were received from the UAMS inpatient pharmacy. The chemotherapeutic agents were reconstituted with sterile saline; stock solutions were stored at 4 °C. PL (Sigma Aldrich) was dissolved in dimethylsulfoxide (DMSO; Fisher) and stored at 4 °C. Animals were divided into 4 treatment groups (*n* = 12 mice/group): Animals were divided into four groups, namely DMSO (control), PL, TAC, and TAC/PL. The TAC and TAC/PL groups each received 4 cycles of weekly doxorubicin (2 mg/kg) and cyclophosphamide (50 mg/kg), followed by 4 cycles of weekly docetaxel (8 mg/kg). The TAC/PL and PL groups received PL (2 mg/kg) weekly (12 weeks) up to 24 h prior to behavioral testing. The control group was injected weekly with 0.5% DMSO for the entire treatment period. All injections were administered intraperitoneally. Sociability behavior testing was conducted 30 days after the last docetaxel injection. Figure 9 shows the graphical representation of the experiment timeline.

### 4.3. Three-Chamber Arena Social Behavior Test

Mice were tested for social memory with an aluminum floor arena, transparent acrylic walls, and an open ceiling (Figure 10). The arena has three adjoining chambers (left, center, and right), each 40 cm × 20 cm × 23 cm in size; each chamber has an opening, connecting the center chamber to the other two. In addition, an aluminum cylindrical cage with plastic ends was placed in the left and the right chambers. The testing was conducted in three 10-min phases per subject. For the first phase (habituation phase), the subject was placed inside the arena in the center chamber and allowed to freely familiarize themselves with the arena (Figure 10A). During phase 2 (sociability phase), a new mouse (all mice are of the same sex, age, and strain) known as “stimulus 1” (stranger 1) was introduced into one of the cylindrical cages (Figure 10B). Finally, during phase 3 (social novelty phase), a second, newer mouse known as “stimulus 2” (stranger 2) was placed in the remaining cylindrical cage. Location bias for phases 2 and 3 was eliminated by randomly selecting left or right chamber per subject (Figure 10C). Mice serving as stimulus 1 and stimulus 2 were chosen randomly and nonconsecutively and were naïve non-aggressive animals that had no contact with testing subjects before the experiment. Animals were sacrificed by cervical dislocation 24 h after behavior testing.

### 4.4. Tissue Preparation for Proteomics Analysis 

The hippocampus was removed from the left hemisphere and placed in 400 µL of RIPA lysis buffer (10 mM Tris-Cl pH 8.0, 1 mM EDTA, 0.5 M EGTA, 1% Triton X-100, 0.1% sodium deoxycholate, 0.1% SDS, 140 mM sodium chloride). The tissue was homogenized on ice, incubated for 30 min, and centrifuged. The supernatant was stored at –80 °C. The Compat-Able Protein Assay Preparation Reagent Kit (Thermo Scientific) was used to eliminate EGTA as an interfering substance for the Pierce BCA Protein Assay Kit (Thermo Scientific). Proteins were separated with SDS-PAGE on 4–15% Criterion TGX Precast Midi Protein Gels (Bio-Rad). Gels were stained with Coomassie blue (Bio-Rad) before being sent to the UAMS proteomics core for mass spectrometry analysis. 

### 4.5. GeLC-MS/MS Analysis

Gel lanes for each sample were cut into 12 equal slices, de-stained in a solution of 50% methanol (Fisher) and 100 mM ammonium bicarbonate (Sigma), followed by reduction in 10 mM Tris[2-carboxyethyl] phosphine (Pierce) and alkylation in 50 mM of iodoacetamide (Sigma). Gel slices were dehydrated in acetonitrile (Fisher), followed by the addition of 100 ng porcine sequencing-grade modified trypsin (Promega) in 100 mM ammonium bicarbonate (Sigma) and incubation at 37 °C for 12–16 h. Peptide products were then acidified in 0.1% formic acid (Pierce). Tryptic peptides were separated by reverse-phase XSelect CSH C18 2.5 µm resin (Waters) on an in-line 150 × 0.075-mm column, using a nanoAcquity UPLC system (Waters). Peptides were eluted over a 30-min gradient from 97:3 to 67:33 buffer A:B ratio (buffer A = 0.1% formic acid, 0.5% acetonitrile; buffer B = 0.1% formic acid, 99.9% acetonitrile). Eluted peptides were ionized by electrospray (2.15 kV), followed by MS/MS analysis using higher-energy collisional dissociation (HCD) on an Orbitrap Fusion Tribrid mass spectrometer (Thermo Fisher) in top-speed data-dependent mode. MS data were acquired with the FTMS analyzer in profile mode at a resolution of 240,000 over a range of 375 to 1500 m/z. After HCD activation, MS/MS data were acquired with the ion trap analyzer in centroid mode and normal mass range with a precursor mass-dependent normalized collision energy between 28.0 and 31.0.

### 4.6. RNA Extraction and Quantitative Reverse Transcription Polymerase Chain Reaction (qRT-PCR) 

The right hemispheres of hippocampi were dissected from each treatment group, immediately frozen in liquid nitrogen, and stored at –80 °C until further processing. Total RNA was extracted from hippocampal tissue with the AllPrep DNA/RNA extraction kit (Qiagen, Valencia, CA, USA), according to the manufacturer’s protocol. RNA quality and quantity were assessed on a Nanodrop 2000 instrument (Thermo Scientific). cDNA was synthesized with random primers and a high-capacity cDNA reverse transcription kit (Applied Biosystems), according to the manufacturer’s protocol (Life Technologies). The levels of gene transcripts were determined by qRT-PCR with TaqMan Gene Expression Assays (Life Technologies and Integrated DNA Technologies), according to the manufacturer’s protocol. In all cases, GAPDH was used as an internal reference gene, and fold changes were calculated with the 2-ddCt method. Measurements were taken in duplicate.

### 4.7. Data Analysis

With MaxQuant version 1.6.5.0 (Max Planck Institute), the UniprotKB database was used to identify and quantify proteins restricted to Mus musculus. The database search parameters included selecting MS1 reporter type, trypsin digestion with up to 2 missed cleavages, fixed modifications for carbamidomethyl of cysteine, variable modifications for oxidation on methionine and acetyl on N-terminus, 5 ppm precursor ion tolerance for the first search and 3 ppm for the main search, and label-free quantitation with iBAQ intensities. A false discovery rate (FDR) threshold of 1% was considered acceptable for identifying protein and peptides. Protein probabilities were assigned by the Protein Prophet algorithm [64]. 

MaxQuant iBAQ intensities for each sample were median-normalized so that the medians were equal to the sample with the maximum median. Median-normalized iBAQ intensities were then imported into Perseus version 1.6.1.3 (Max Planck Institute) to perform log2 transformation and impute the missing values, using a normal distribution with a width of 0.3 and a downshift of 2 standard deviations. The linear models for the microarray data (Limma) Bioconductor package were used to calculate differential expression among the treatment conditions, using lmFit and eBayes functions [65]. Proteins with a fold change >2 and FDR-adjusted *p* < 0.05 were considered significantly different. Ingenuity Pathway Analysis (IPA) (Qiagen) was used to conduct pathway and network analysis of differentially expressed proteins.

Data are expressed as means ± standard error of the mean (SEM). Statistical analysis was conducted with Graphpad Prism 8.0 software (Graphpad), and *p* < 0.05 was considered statistically significant. The results of the sociability behavior test were analyzed with one-way ANOVA.

## 5. Conclusions

To conclude, our examination of the ability of PL to act as a neuroprotectant against chemotherapy-induced cognitive impairment revealed that PL does protect against TAC-induced social memory deficits. Mice treated with TAC had impaired social memory, but mice co-treated with TAC/PL did not. Our proteomics analysis indicates a complex array of pathways through which this process may be possible. Although PL reportedly has antioxidant properties, all our treatment groups showed reduced mRNA expression of Nrf2, the master regulator of many antioxidant genes. The antioxidant genes examined in this study displayed expression effects that contradicted those of the commonly known Nrf2/ARE pathway, suggesting that PL might exert neuroprotection through other pathways. The mechanism of PL neuroprotection action has not yet been elucidated; however, as a known senolytic and anti-inflammatory agent, PL might be acting as a neuroprotectant against TAC via mechanisms that involve senescence or neuroinflammation, which were not pursued in this study. Our proteomics analysis indicates a complex array of pathways implicated in TAC/PL neuroprotection. TAC had no effect on expression levels of genes encoding NMDA subunits, but it increased expression of GluA2, leading us to speculate that AMPA receptors are more involved in protecting social memory. Future examination of other pathways involved in chemotherapy-induced cognitive impairment, and a full understanding of other mechanisms of PL action, will provide a clear picture of how PL is neuroprotective against the effects of TAC.

## Figures and Tables

**Figure 1 ijms-23-02008-f001:**
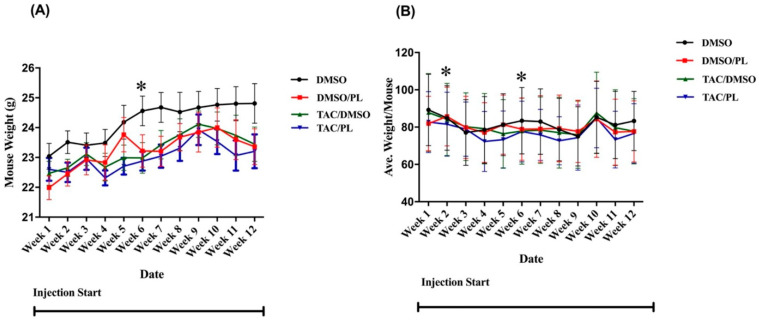
Mouse weights and food consumption during the injection period. (**A**) Mouse weights were significantly different between other treatment groups and control-treated animals in week 6. (**B**) Food consumption was significantly different in weeks 2 and 6. Two-way ANOVA, average ± SEM (*n* = 12); Bonferroni multiple comparisons * *p* < 0.05.

**Figure 2 ijms-23-02008-f002:**
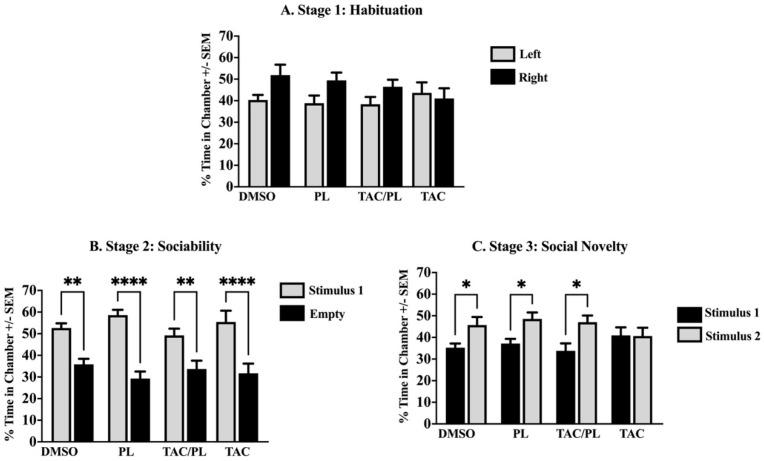
Sociability. (**A**) No significant difference in chamber exploration time in all groups. (**B**) Ordinary social behavior in all treatment groups by spending significant time in a chamber with the novel mouse (stimulus 1). (**C**) Animals treated with DMSO, PL, and TAC/PL distinguished the previously present mouse (stimulus 1) from the newly presented mouse (stimulus 2). Only the TAC-treated animals failed to distinguish between the previously present mice (stimulus 1) and the newly introduced mice (stimulus 2). One-way ANOVA, average ± SEM; Holm-Sidak multiple comparisons * *p* < 0.05, ** *p* < 0.01, **** *p* < 0.0001. (*n* = 12).

**Figure 3 ijms-23-02008-f003:**
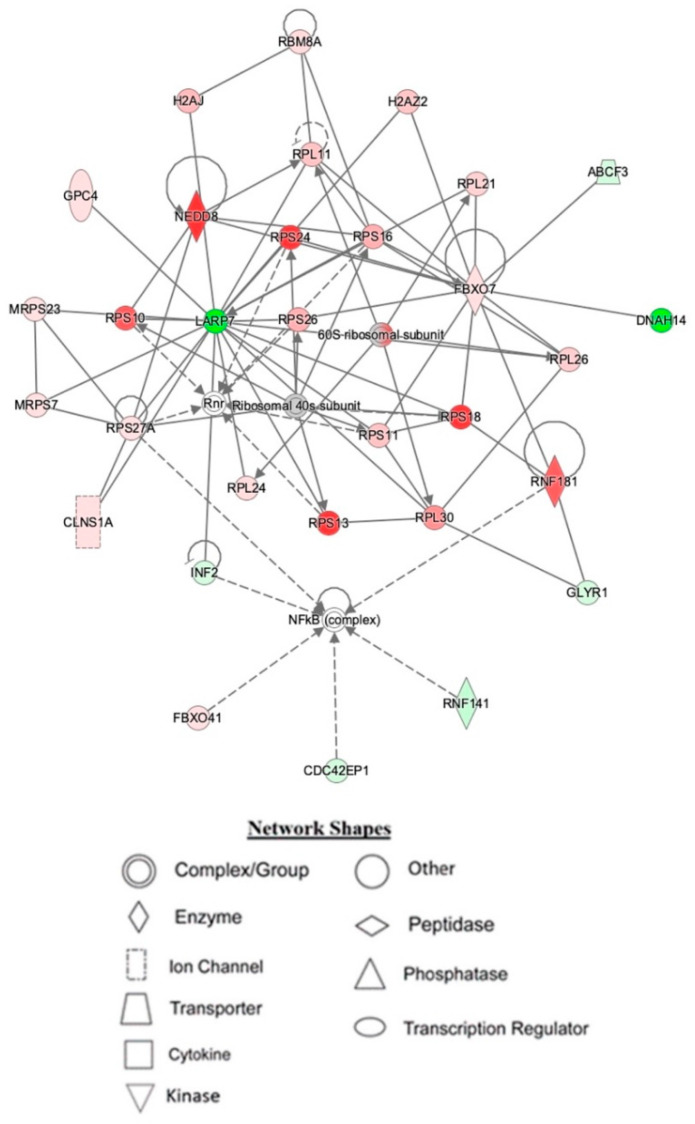
Depiction of mouse hippocampal network 1, affected by IPA resulting from TAC treatment. Node color is indicative of differential expression. Red represents upregulation, and green represents downregulation. Color intensity directly corresponds to the degree of regulation. Gray nodes represent proteins found in the data set but were not significantly expressed. Uncolored nodes represent proteins not differentially expressed but were incorporated into the computational network based on evidence stored in the Ingenuity Knowledge Base. The arrows and blocked lines indicate known direct and indirect interaction as well as the direction of the interaction.

**Figure 4 ijms-23-02008-f004:**
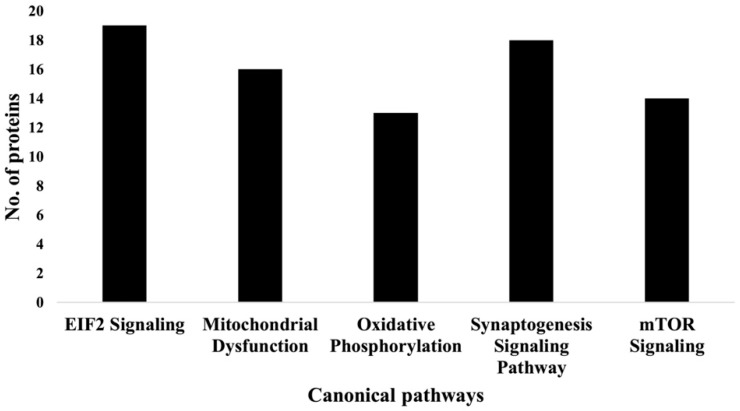
Graph representation of the numbers of molecules identified in the top 5 canonical pathways affected by TAC treatment.

**Figure 5 ijms-23-02008-f005:**
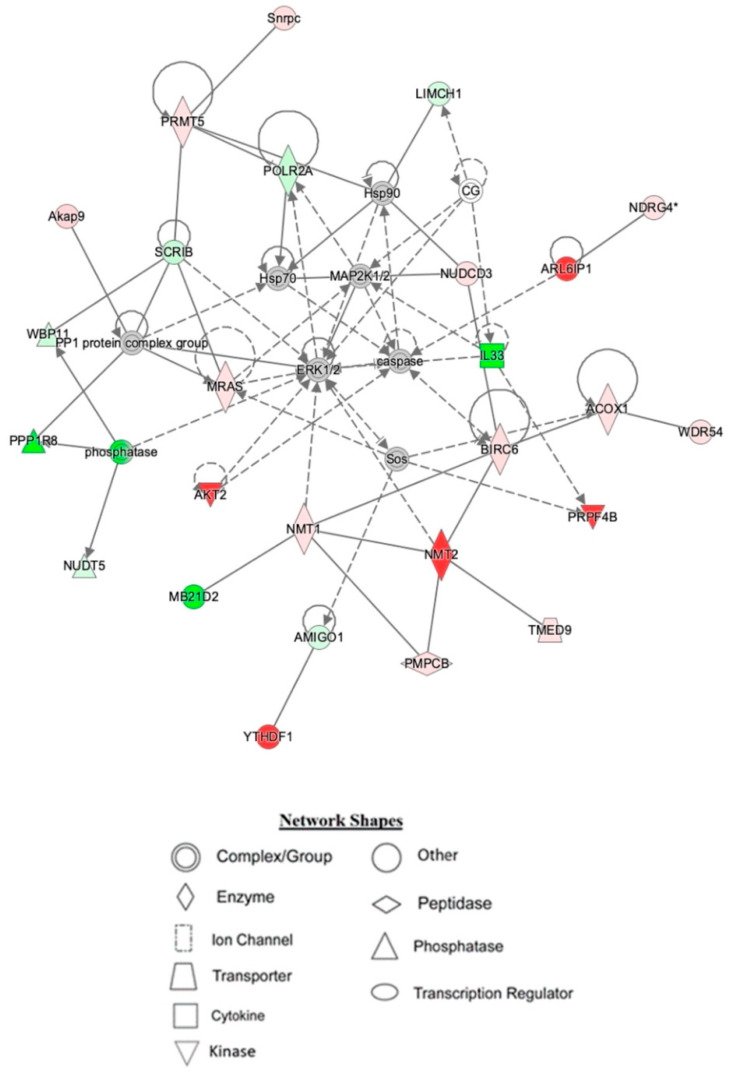
Depiction of hippocampal network 1 identified by IPA being affected by TAC/PL co-treatment to TAC treatment. Node color is indicative of differential expression. Red represents upregulation, and green represents downregulation. Color intensity directly corresponds to the degree of regulation. Gray nodes represent proteins found in the data set but were not significantly expressed. Uncolored nodes represent proteins not differentially expressed but were incorporated into the computational network based on evidence stored in the Ingenuity Knowledge Base. The arrows and blocked lines indicate known direct and indirect interaction as well as the direction of the interaction.

**Figure 6 ijms-23-02008-f006:**
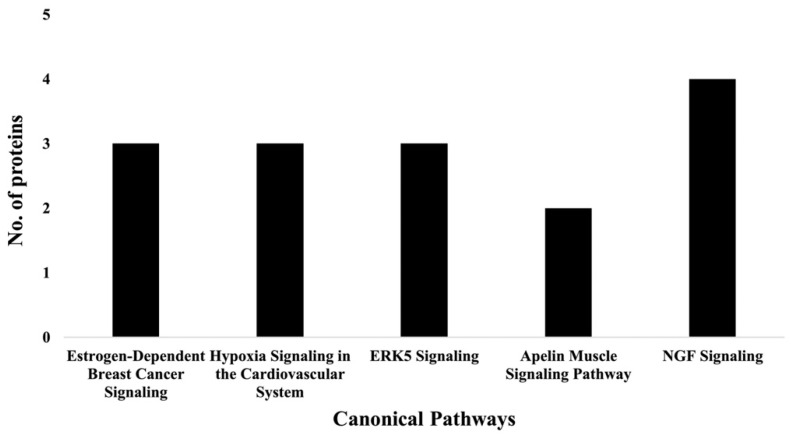
Graph representation of the number of molecules identified in the top 5 canonical pathways affected by PL (i.e., TAC/PL treatment compared to TAC treatment).

**Figure 7 ijms-23-02008-f007:**
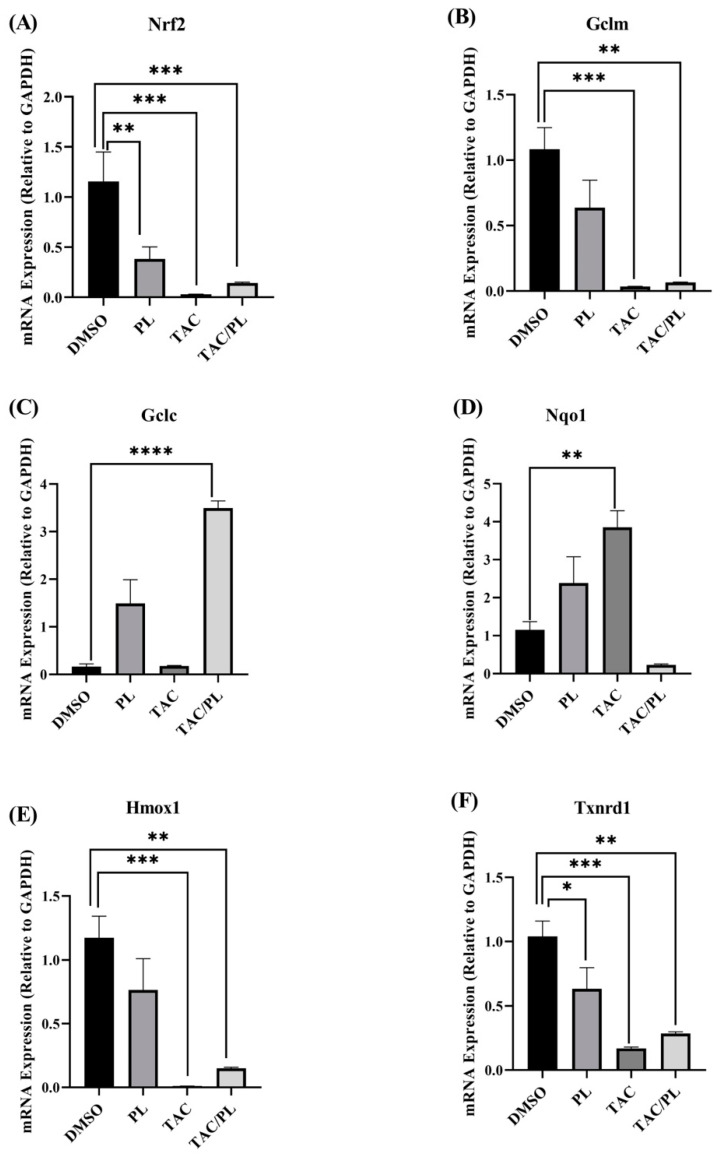
Changes in mRNA expression of Nrf2 cytoprotective target genes. (**A**) PL, TAC, and TAC/PL treatments reduced expression of Nrf2. (**B**) PL, TAC, and TAC/PL treatments reduced expression of GCLM. (**C**) TAC and TAC/PL treatments increased expression of GCLC. (**D**) TAC treatment increased expression of Nqo1. (**E**) TAC and TAC/PL treatments reduced expression of Hmox1. (**F**) PL, TAC, and TAC/PL treatments reduced expression of Txnrd1. One-way ANOVA, average ± SEM; Bonferroni multiple comparisons * *p* < 0.05, ** *p* < 0.01, *** *p* < 0.001, **** *p* < 0.0001. DMSO, *n* = 7; PL, *n* = 9; TAC, *n* = 8; TAC/PL, *n* = 6.

**Figure 8 ijms-23-02008-f008:**
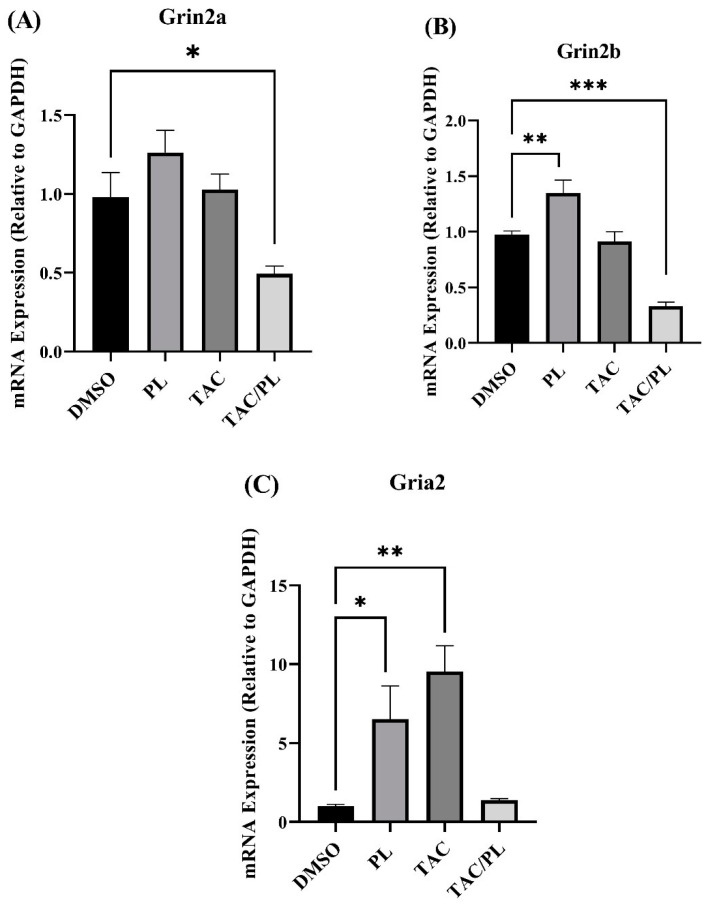
Changes in mRNA levels of genes encoding NMDA and AMPA receptor subunits (**A**) TAC/PL treatment reduced mRNA levels of Grin2a. (**B**) PL treatment increased and TAC/PL treatment reduced mRNA levels of Grin2b. (**C**) PL and TAC treatments increased mRNA levels of Gria2. One-way ANOVA, average ± SEM; Bonferroni multiple comparisons * *p* < 0.05, ** *p* < 0.01, *** *p* < 0.001. DMSO *n* = 7; PL *n* = 9; TAC *n* = 8; TAC/PL *n* = 6.

**Figure 9 ijms-23-02008-f009:**
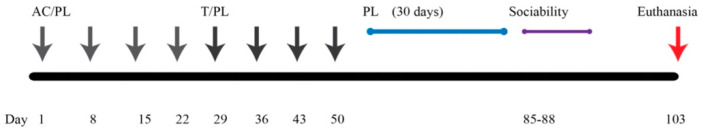
Graphical representation of experimental design and timeline.

**Figure 10 ijms-23-02008-f010:**
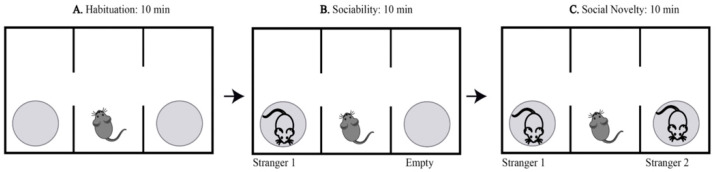
Graphical representation of three-chambered social behavior test. The habituation phase of three-chamber sociability testing, the test subject is allowed to familiarize with arena for 10 min (**A**). Sociability testing phase of three-chamber sociability testing, a stranger mouse is introduced for 10 min (**B**). Social novelty phase of three-chamber sociality testing, a newer stranger and familiar mouse are both present for 10 min (**C**).

**Table 1 ijms-23-02008-t001:** Top upregulated and downregulated proteins associated with TAC treatment compared to DMSO treatment.

Protein	Symbol	Description	Location	Type	Fold Change (Log Ratio)
Q548W7	DBI	Diazepam Binding Inhibitor, Acyl-CoA Binding Protein	Cytoplasm	Other	17.06
Q56A15	CYCS	Cytochrome C, Somatic	Cytoplasm	Transporter	16.93
Q54A87	ATP6V1G2	ATPase H+ Transporting V1 Subunit G2	Cytoplasm	Transporter	15.90
Q91WS0	CISD1	Cdgsh Iron Sulfur Domain 1	Cytoplasm	Other	15.29
Q9D6H6	NDUFB3	NADH: Ubiquinone Oxidoreductase Subunit B3	Cytoplasm	Enzyme	13.69
A0A571BGC3	DNAH14	Dynein Axonemal Heavy Chain 14	Other	Other	−11.80
A2AWN8	YTHDF1	Yth N6-Methyladenosine RNA Binding Protein 1	Other	Other	−10.52
A2AJH3	NMT2	N-Myristoyltransferase 2	Cytoplasm	Enzyme	−10.48
Q61136	PRPF4B	Pre-MRNA Processing Factor 4B	Nucleus	Kinase	−9.92
I3PQW3	LARP7	La Ribonucleoprotein 7, Transcriptional Regulator	Nucleus	Other	−9.32

**Table 2 ijms-23-02008-t002:** Top upregulated and downregulated proteins associated with TAC-treated mice compared to TAC/PL-treated mice.

Protein	Symbol	Description	Location	Type	Fold Change (Log Ratio)
B0LAE4	ARL6IP1	ADP Ribosylation Factor Like GTPase 6 Interacting Protein 1	Cytoplasm	Other	11.11
Q3UMB9	WASHC4	Wash Complex Subunit 4	Cytoplasm	Other	9.87
O35075	VPS26C	Vps26 Endosomal Protein Sorting Factor C	Nucleus	Other	9.64
A2AWN8	YTHDF1	Yth N6-Methyladenosine RNA Binding Protein 1	Other	Other	9.62
F8WHG5	AKT2	Akt Serine/Threonine Kinase 2	Cytoplasm	Kinase	9.58
A0A0R4J0T5	Celf1	Cugbp, Elav-Like Family Member 1	Nucleus	Other	−10.71
D3Z742	MB21D2	Mab-21 Domain Containing 2	Other	Other	−10.42
A2ADR8	PPP1R8	Protein Phosphatase 1 Regulatory Subunit 8	Nucleus	Phosphatase	−10.34
Q99JT1	GATB	Glutamyl-TRNA Amidotransferase Subunit B	Cytoplasm	Enzyme	−9.90
Q9D0K0	TBC1D7	Tbc1 Domain Family Member 7	Cytoplasm	Other	−9.55

**Table 3 ijms-23-02008-t003:** Top 5 IPA protein networks associated with TAC treatment.

Network Rank	Network Description
1	Associated network functions: RNA damage and repair, protein synthesis, cancer Number of “focus molecules” contained in network: 31 IPA *p*-score: 61 Network Proteins: 60S ribosomal subunit, ABCF3, CDC42EP1, CLNS1A, DNAH14, FBXO41, FBXO7, GLYR1, GPC4, H2AJ, H2AZ2, INF2, LARP7, MRPS23, MRPS7, NEDD8, NFkB (complex), RBM8A, RNF141, RNF181, RPL11, RPL21, RPL24, RPL26, RPL30, RPS10, RPS11, RPS13, RPS16, RPS18, RPS24, RPS26, RPS27A, Ribosomal 40s subunit, Rnr
2	Associated network functions: dermatological diseases and conditions, hair and skin development and function, organ development Number of “focus molecules” contained in network: 26 IPA *p*-score: 47 Network proteins: 26s Proteasome, AGA, ATP6V1G2, Alp, BCS1L, COPS8, CTSA, DCTN6, DNAJB5, DYNLL2, FBXO3, HPCAL1, Hsp70, Hsp90, IGHG1, Ikb, MERTK, MRPS10, MYCBP2, MYL6, NUDCD3, Nos, P38 MAPK, PSMD4, PTGES3, SKP1, SOD1, SSR4, SUMO, TIAL1, Tmsb4x (includes others), UBE2M, UBR4, Ubiquitin, VPS33B
3	Associated network functions: energy production, nucleic acid metabolism, small-molecule biochemistry Number of “focus molecules” contained in network: 25IPA *p*-score: 44 Network proteins: ANAPC1, ARL8B, ATP5F1D, ATP5MD, ATP5MF, ATP5MG, Atp5k, CISD1, CK1, COTL1, CRELD1, Calcineurin protein(s), E3 RING, ERK, FKBP1A, MTFP1, NDUFA4, NRBP1, OTUD7A, PPIA, PPIB, PRDX5, SERPINA3, SLC4A1, SPCS2, SPCS3, TCF, TH2 Cytokine, TOMM22, TSC22D1, VHL, adenosine-tetraphosphatase, chymotrypsin, peptidylprolyl isomerase, trypsin
4	Associated network functions: cellular assembly and organization, behavior, cellular compromise Number of “focus molecules” contained in network: 23IPA *p*-score: 40 Network proteins: ADCY1, ASAH1, CFL1, CHGB, Cofilin, Cyclin E, DSTN, ENSA, ERK1/2, FTH1, Ferritin, GLRX3, GPIIBIIIA, ISCU, ITGB1BP1, LYPD1, Lfa1, NF1, NMT2, NPC2, Ngf, PCSK1N, PMPCB, PRUNE2, RABGEF1, RAP2A, RAP2B, RTN4IP1, Rap, Rock, Rsk, TLN1, TSH, VGF, c-Src
5	Associated network functions: cell signaling, molecular transport, nucleic acid metabolism Number of “focus molecules” contained in network: 20IPA *p*-score: 33 Network proteins: 14-3-3, ACBD5, AKTIP, ATP6AP2, AURK, Akt, Alpha tubulin, Arp2/3, BETA TUBULIN, CA1, CDK4/6, Ck2, DYNLL1, DYNLT3, Dynein, E2f, GMFB, GOLGA4, GPX1, HPCAL4, INPP1, NFkB (family), NOP58, PLXDC2, RABL3, RCN1, RHOG, SNCA, STMN1, Synuclein, TUBA8, UQCR10, Vdac, glutathione peroxidase, tubulin

Qiagen IPA algorithm overlays focus molecules from the experimental dataset to the Global Molecular Network and generates a connectivity map. The *p*-score (log 10 (*p*-value) is calculated by Fisher’s exact test and is indicative of the probability of focus molecules in a network being selected randomly from the Global Molecular Network.

**Table 4 ijms-23-02008-t004:** Top 5 canonical pathways affected by TAC treatment, as highlighted by IPA.

Pathway Name	*p*-Value	IPA Ratio
EIF2 signaling	1.000 × 10^−11^	0.0848
Mitochondrial dysfunction	1.000 × 10^−10^	0.0936
Oxidative phosphorylation	2.818 × 10^−10^	0.1190
Synaptogenesis signaling pathway	1.622 × 10^−08^	0.0577
mTOR signaling	1.122 × 10^−07^	0.0667

The IPA ratio is the number of molecules that meet criteria divided by the total number of pathway proteins in the IPA database. The *p*-value represents the probability of the ratio occurring by chance.

**Table 5 ijms-23-02008-t005:** Top 5 networks associated with TAC/PL vs. TAC treatment.

Network Rank	Molecules in Network
1	Associated network functions: post-translational modification, RNA post-transcriptional modification, cellular development Number of focus molecules in network: 26 IPA *p*-score: 61 Network proteins: ACOX1, AKT2, AMIGO1, ARL6IP1, Akap9, BIRC6, CG, ERK1/2, Hsp70, Hsp90, IL33, LIMCH1, MAP2K1/2, MB21D2, MRAS, NDRG4, NMT1, NMT2, NUDCD3, NUDT5, PMPCB, POLR2A, PP1 protein complex group, PPP1R8, PRMT5, PRPF4B, SCRIB, Snrpc, Sos, TMED9, WBP11, WDR54, YTHDF1, caspase, phosphatase
2	Associated network functions: cellular function and maintenance, nervous system development and function, tissue development Number of focus molecules in network: 19 IPA *p*-score: 40 Network proteins: 26s proteasome, ANKRD17, ATP5F1C, Akt, Ap1, BLVRB, CD3, CREB1, Calmodulin, Celf1, DDX46, ERK, GPR37, Histone h3, IGKC, Immunoglobulin, Insulin, LARP7, ME1, NFkB (complex), *p*-TEFb, P38 MAPK, PI3K (complex), PNN, PSMD3, RAS, RBMX, RNA polymerase II, SCAF8, SCARB2, TBC1D7, TCR, TRPM3, TUB, UBR4
3	Associated network functions: cellular assembly and organization, cell morphology, cellular function, and maintenance Number of focus molecules in network: 16 IPA *p*-score: 32 Network proteins: ASB13, BAZ1B, CDC42BPA, CHST11, DCAKD, EPG5, FSCN2, GPR161, IRF2BPL, KLHL28, MFSD11, NCLN, PARD3B, PHACTR4, PHF3, RHOBTB2, RNF20, SPECC1L, SPRYD7, SYN3, TIMM23, TRIM25, UBA6, UBC, UBE2Q1, URM1, USP32, USP45, USP9Y, VCP, VIRMA, VPS35L, WASHC4, WASHC5, ZMIZ2
4	Associated network functions: drug metabolism, endocrine system development and function, lipid metabolism Number of focus molecules in network: 13 IPA *p*-score: 25 Network proteins: ABHD4, ADGRL1, ANXA4, AP1M1, ARHGAP6, CCNT2, DFFB, HNRNPL, HRH3, HSD17B6, HTR4, ITIH4, JOSD2, NEK4, OMG, PCDHGB1, PDS5A, POMC, PRR5, PRUNE2, PTBP2, RAMP3, RBM10, SMC3, SPECC1, SRP54, STK32C, TATDN1, TBC1D5, TECTA, TENM2, VPS26C, VPS35L, ZFP64, beta-estradiol
5	Associated network functions: protein synthesis, cellular movement, hematological system development and function Number of focus molecules in network: 11 IPA *p*-score 20 Network proteins: ACY1, AKT1S1, ALDH5A1, APP, CPQ, CPT1C, D2HGDH, DGLUCY, DNAH1, DNAH14, DNAH2, DNALI1, ENO3, BXO3, FGF6, GATB, HCAR2, ITIH1, Ighg2b, LGMN, LMCD1, MBD4, METAP2, MRPS6, MTERF3, ROBO3, RPS6KC1, SHFL, SLAMF1, SPHK1, ST3GAL5, TGFB1, TRIM63, ZHX2, ceramide

Qiagen IPA algorithm overlays focus molecules from the experimental dataset to the Global Molecular Network and generates a connectivity map. The *p*-score (log 10 (*p*-value) is calculated by Fisher’s exact test and is indicative of the probability of focus molecules in a network being selected randomly from the Global Molecular Network.

**Table 6 ijms-23-02008-t006:** Top 5 canonical pathways affected by TAC/PL treatment compared to TAC alone, as highlighted by IPA.

Canonical Pathways	*p*-Value	IPA Ratio
NGF signaling	8.913 × 10^−4^	0.0351
Apelin muscle signaling pathway	2.291 × 10^−3^	0.1050
ERK5 signaling	2.512 × 10^−3^	0.0417
Estrogen-dependent breast cancer signaling	2.692 × 10^−3^	0.0405
Hypoxia signaling in the cardiovascular system	2.692 × 10^−3^	0.0405

The IPA ratio is the number of molecules that meet criteria divided by the total number of pathway proteins in the IPA database. The *p*-value represents the probability of the ratio occurring by chance.

## Data Availability

The mass spectrometry proteomics data have been deposited to the ProteomeXchange Consortium via the PRIDE [1] partner repository, but we have not received the dataset identifier.

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
