# Peer review of "Piperlongumine as a Neuro-Protectant in Chemotherapy Induced Cognitive Impairment"

_ijms, 2022, doi:10.3390/ijms23042008_

Round 1

Reviewer 1 Report

The manuscript by Ntagwabira et al. addresses a significant issue linked with chemotherapy, which is cognitive impairment following treatment. The research examines the potential use of piperlongumine, an alkaloid with senolytic, anti-inflammatory and selective cytotoxic properties, as a neuroprotective agent against chemotherapy-induced cognitive impairment associated with the docetaxel-doxorubicin-cyclophosphamide (TAC) regimen.

Minor comments:

  1. The introduction should be checked for mistakes or missing words, for instance: line 40: ‘The need therapies that can prevent or alleviate these side effects is thus crucial’, line 77: ‘improves cognition in an and aged mice’.
  2. There are some unnecessary spaces throughout the entire manuscript: lines 55, 66, 77, 78, 110.
  3. Other minor mistakes include: line 97: ‘in the weeks of week 2 and week 6’, line 194: ‘indicate a significant decreases’, line 197: ‘a decrease from DMSO to PL (p<0.05), DMSO to PL (p<0.001)’, line 210: ‘PL and TAC treatment decreased Gria2 mRNA levels’ (the graph shows the opposite), line 215: ‘A In the current study’.

The manuscript is complex, well written, the research methods chosen for this study are adequate, the results are well presented and discussed, and the work lies within a topic of high current interest. Consequently, it is my opinion that this manuscript will be suitable for publication in International Journal of Molecular Sciences, subject to minor revisions.

Author Response

The entire manuscript was checked and corrected for spelling and punctuation mistakes, missing words, and unnecessary spaces. Line 252 ‘PL and TAC treatment increased Gria2 mRNA levels’ was corrected to reflect the graph 8 (C). Missing words in line 80 were added and the references Go et al. were corrected in text and reference list.

Reviewer 2 Report

In the present manuscript Ntagwabira et al. investigated the neuroprotective effect of piperlongumine on cognitive impairment during chemiotherapy. Overall, the manuscript is well designed and written. The topic is extremely interesting and of great relevance.

Author Response

Per referees’ recommendations, the entire manuscript was checked and corrected for spelling and punctuation mistakes, missing words, and unnecessary spaces. Line 252 ‘PL and TAC treatment increased Gria2 mRNA levels’ was corrected to reflect the graph 8 (C). Missing words in line 80 were added and the references Go et al. were corrected in text and reference list.